# Non-Surgical Management of the Gingival Smile with Botulinum Toxin A—A Systematic Review and Meta-Analysis

**DOI:** 10.3390/jcm12041433

**Published:** 2023-02-10

**Authors:** Carolina Rojo-Sanchis, José María Montiel-Company, Beatriz Tarazona-Álvarez, Orion Luiz Haas-Junior, María Aurora Peiró-Guijarro, Vanessa Paredes-Gallardo, Raquel Guijarro-Martínez

**Affiliations:** 1Department of Orthodontics, Universidad de Valencia, 46010 Valencia, Spain; 2Department of Oral and Maxillofacial Surgery, São Lucas Hospital of PUCRS, Pontifical Catholic University of Rio Grande do Sul, Porto Alegre 90619-900, Brazil; 3Department of Orthodontics, Cardenal Herrera-CEU Universidad de Valencia, 46115 Valencia, Spain

**Keywords:** gummy smile, excessive gingival display, hyperfunction, lip elevator, BTX-A, botulinum toxin

## Abstract

Currently, concern about facial attractiveness is increasing, and this fact has led to orthodontics in adult patients being an increasingly demanded treatment, and with it, multi-disciplinary work. When it is caused by a vertical excess of the maxilla, the ideal solution is orthognathic surgery. However, in borderline cases and when the cause is hyperactivity of the upper lip levator muscle complex, alternative conservative solutions can be considered, such as the application of botulinum toxin A (BTX-A). Botulinum toxin is a protein produced by a bacterium and causes a reduction in the force of muscle contraction. The multi-factorial nature of the smile requires an individualized diagnosis in each patient, since there are multiple ways to treat the gummy smile (orthognathic surgery, gingivoplasty, orthodontic intrusion). In recent years, interest has grown in the simplest techniques that allow the patient to quickly return to their usual routine, such as lip replacement. However, this procedure shows recurrences in the first 6–8 post-operative weeks. The main objective of this systematic review and meta-analysis is to analyze the effectiveness of BTX-A in the treatment of gummy smile in the short term, to study its stability, and to evaluate potential complications. A thorough search of the PubMed, Scopus, Embase, Web of Science, and Cochrane databases and a grey literature search were conducted. The inclusion criteria were studies with a sample size greater than or equal to 10 patients with gingival exposure greater than 2 mm in smile, treated with BTX-A infiltration. Those patients whose exclusive etiology of their gummy smile was related to altered passive eruption, gingival thickening, or overeruption of upper incisors were excluded. In the qualitative analysis, the mean pre-treatment gingival exposure ranged between 3.5 and 7.2 mm, reaching a reduction of up to 6 mm after infiltration with botulinum toxin at 12 weeks. Although multiple muscles are involved in the facial expression, the muscles par excellence selected for blockade with BTX-A were levator labii superioris, levator labii superioris ala nasalis, and zygomaticus minor, infiltrating from 1.25 to 7.5 units per side. In the quantitative analysis, the difference in mean reduction between both groups was −2.51 mm at two weeks and −2.24 mm at three months. The benefit of BTX-A in terms of improvement of gummy smile is demonstrated, as a significant reduction in gummy smile is estimated by BTX-A therapy two weeks after its application. Its results gradually decrease over time, however, they stay satisfactory without returning to their initial values after 12 weeks.

## 1. Introduction

Currently, concern about facial attractiveness is increasing, becoming a key element in the social life of a large part of the population. This fact has led to orthodontics in adult patients being an increasingly demanded treatment, and with it aesthetic excellence and multi-disciplinary work.

The smile is probably the most pleasant human expression, the result of an interaction of three components (dental, labial, and gingival), which is produced thanks to the contraction of certain muscles located in the middle and lower thirds of the face [1,2]. It is considered the key to expression in human social life that allows people to transmit emotions and personality, in addition to being considered a fundamental work tool. It is the central point of dental work and for all this, it is a relevant aesthetic criterion to study in patients asking for improvement in this facial expression [3].

Increased gingival exposure is a common finding in the population, with a prevalence rate ranging between 10% and 29% [4], being more frequent in women [5]. Its etiology can be attributed to multiple gingival factors (altered passive eruption or gingival enlargement due to drugs), skeletal (maxillary vertical excess), muscular (hypermobility of the upper lip), or anatomical (length of the upper lip or clinical crown, alveolar extrusion or dentoalveolar) [6,7].

The multi-factorial nature of the smile requires an individualized diagnosis in each patient, since there are multiple ways to treat the gummy smile (orthognathic surgery, gingivoplasty, orthodontic intrusion) [1,5,6,7,8,9]. In recent years, interest has grown in the simplest techniques that allow the patient to quickly return to their usual routine, such as lip replacement. However, this procedure shows recurrences in the first 6–8 post-operative weeks [10]. That is why the infiltration of botulinum toxin type A (BTX-A) has gained importance as a non-surgical, effective, and minimally invasive alternative for the management of the gummy smile. Anatomical factors, such as the length of the lip, must be diagnosed before planning BTX-A therapy to avoid subsequent complications [11,12,13].

Botulinum toxin is a protein produced by the bacterium Clostridium botulinum. It inhibits the release of the neurotransmitter acetylcholine, which causes a reduction in the force of muscle contraction. Its clinical effect has an average duration of between 4 and 6 months. In the context of gummy smile treatment, the main treatment targets are the levator labii superioris, and levator labii superioris and ala nasalis muscles, which are primarily responsible for central lip elevation and are located at more accessible and safer injection sites [1,13,14,15]. It is a simple technique with a low rate of complications, including injection site pain, edema, hematoma, mild erythema, and asymmetric smile [5,16]. In the perioral area, being highly mobile, the duration of BTX-A may be reduced when compared to other areas with less mobility in the upper facial third [5,17].

Unlike BTX-A, hyaluronic acid is a highly viscous mucopolysaccharide found naturally in the skin and other body tissues; its osmotic capacity gives it moisturizing and volumizing properties [18,19]. In the perioral region, as it is an area of high mobility and where very dense hyaluronic agents cannot be used, its duration ranges between 10 and 18 months [18,19,20].

The management of the gummy smile through the use of fillers is increasing. Specifically, the work by Diaspro et al. in 2018 [10] involved the infiltration of hyaluronic acid in the piriformis opening, achieving an average reduction of 1.37 mm in gingival exposure at two weeks. Recently, another study referred to the use of this filler material, reducing muscle contraction by adding weight to the corresponding soft tissue [21].

Despite its advantages and its widespread use, there is still no consensus in the literature regarding the use of BTX-A.

The objective of this study is to estimate the improvement in the gummy smile with BTX-A in the short term, analyzing its stability up to three months of post-treatment follow-up. In addition, the decrease in gingival exposure after infiltration of hyaluronic acid is compared.

## 2. Materials and Methods

### 2.1. Study Design and Registration

The systematic review was conducted in accordance with the preferred reporting items for systematic reviews and meta-analyses guidelines (PRISMA) [22] and was previously registered with PROSPERO under registration number CRD42019115783.

### 2.2. Search of the Literature Process

The objective was to answer the following research question: what effects does BTX-A infiltration produce in patients with a gummy smile when compared before and after the intervention? For the research question, the “PICO” strategy was used, which stands for population (population/participant), intervention (intervention/exposure for observational studies), comparison (comparison), and outcomes (outcome). To identify the potentially relevant studies irrespective of language, a thorough electronic search was made in the PubMed, Scopus, Embase, Web of Science, and Cochrane databases. An electronic search of the grey literature was made through Opengrey. In particular cases the authors of the articles were contacted by email or ResearchGate to request missing information. The reference lists of the studies included were hand-searched to identify and examine articles not found in the databases that might meet the inclusion criteria. This systematic review was updated in June 2021. The search strategy included 5 Medical Subject Heading (Mesh) terms: “Gingiva”, “Smile”, “Smiling”, “Hypermobility”, and “Botulinum toxin” and 13 uncontrolled descriptors: “Excessive gingival display”, “Gum”, “Hyperfunction”, “Hyperfunctional”, “Upper lip”, “Lip elevator”, “Vertical maxillary excess”, “Botox”, “BTX-A”, “OnabotulinumtoxinA”, “Dermal filler”, “AbobotulinumtoxinA”, and “Neuromuscular blocking agent”. Boolean operators (“OR” and “AND”) were used to put together terms (MeSH/non-MeSH) related to the research question. These keywords were divided into two groups: 11 primary keywords related to gummy smile terminology and its etiology, and 7 secondary keywords related to fillers. Searches were made for all the possible combinations between the terms in the two groups, separately and combined. The articles identified were exported to Mendeley Desktop 1.13.3 software (Mendeley Ltd., London, England) to check for duplicates.

### 2.3. Inclusion and Exclusion Criteria

“Articles” and “articles in press” were included. Randomized clinical trials (RCTs), cohort studies, and case-controlled were included. Retrospective and prospective studies were included. Case reports, case series, non-systematic reviews of the literature, and editorials were excluded. No restriction was placed on publication year or language. The selection criteria were: studies with a sample size greater than or equal to 10 adult patients, with gingival exposure greater than 2 mm in a smile with hyperfunction of the upper lip, with or without vertical excess of the maxilla and treated with BTX infiltration. Those patients whose exclusive etiology of their gummy smile was related to altered passive eruption, gingival thickening, or overeruption of upper incisors were excluded.

### 2.4. Data Extraction

Two reviewers (CR-S and MAP-G), working independently, systematically assessed the titles and abstracts of all the articles identified. If they disagreed, a third reviewer (RG-M) was consulted. If the abstract did not contain sufficient information to reach a decision, the reviewers read the full article before taking the final decision. Subsequently, the full texts of all the articles were read and the reasons for rejecting those excluded were recorded. The kappa statistic was used to assess the level of agreement between authors.

*Study data.* The following variables were recorded for each article: author and year of publication, type of study, sample size (including losses during the study), and demographic variables (sex and age). To evaluate the improvement in the gummy smile after treatment, the infiltrated units in each patient and injection area were analyzed.

### 2.5. Risk of Bias

Study quality was analyzed by the same investigators independently (CRS, MAPG), using the Cochrane collaboration’s ROBINS-I tool for nonrandomized studies of interventions (NRSI) [23]. The tool consists of 7 items pre-intervention (confounding and selection bias), intervention (classification bias), and post-intervention (report bias). Studies were judged to be at low risk (low risk of bias for all domains), moderate risk (low or moderate risk of bias for all domains), serious risk (serious risk of bias for at least one domain, but not a critical risk of bias in any domain), critical risk (critical risk of bias in at least one domain), or missing information (there is no clear indication that the study is at serious or critical risk of bias and there is a lack of information in one or more key bias domains).

### 2.6. Data Synthesis and Statistical Analysis

The initial, two weeks and three months post-infiltration with BTX-A means and confidence intervals were recorded for the variable “gingival exposure”. For quantitative synthesis, studies were combined using a random effects model with the inverse variance method. The effect size was determined with the means difference and their confidence intervals between the stages of two weeks and three months post-infiltration and the initial stage. Heterogeneity was assessed with the Q test and the I^2^ test. Heterogeneity was considered when the *p* value of the Q test was less than 0.1. To assess the influence of the units of Botox injected, a meta-regression analysis was performed and a meta-analysis by subgroups was performed to compare the reduction between the measurements at two weeks and three months. Publication bias was assessed using funnel plots, the Duval and Tweedie’s trim and fill method, and the classic fail-safe number value. Comprehensive meta-analysis V 3.0 software was used.

## 3. Results

### 3.1. Results of the Search Process

The search identified 1250 preliminary references related to changes in the gummy smile following BTX-A therapy, of which 236 were found in PubMed, 26 in Scopus, 63 in Cochrane, 506 in Embase, 362 in Web of Science, 53 in the grey literature search (OpenGrey), and 4 through hand-searching based on the references cited in the articles included. After excluding 508, the remaining 742 were screened. Of these, 185 were excluded on reading the title and abstract as they were unrelated to the research question. After examining the full text of the resulting 49 articles, 34 were excluded for the following reasons: 17 did not answer the PICO question, 11 were reviews of the literature, narratives, or letters to editor, and 6 had a sample size of fewer than 10 patients. Finally, 15 articles met the inclusion criteria and were included in the qualitative review, and 11 were included in the quantitative review (meta-analysis). The level of agreement between investigators was κ = 0.74 (95% CI, 0.56–0.93). The PRISMA flowchart [24] (Figure 1) provides an overview of the item selection process.

Risk of bias assessment of the included studies using the ROBINS tool did not show a high risk of bias in any of the studies, however, most studies (*n* = 12, 92.3%) had moderate risk of bias. Moderate risk of bias was reported for study participant selection, classification of interventions, deviations from intended interventions, missing data, outcome measurements, and selection of reported outcomes in most studies. Only one study with low risk of bias was found [25].

Of the 15 studies, 14 were prospective and 1 retrospective [12]. All studies were controlled clinical trials. The selected patients were of adult age, diagnosed with a gummy smile with an exposure greater than 2 mm, and treated with BTX-A infiltration with or without previous or additional treatments, and were provided with pre- and post-treatment.

### 3.2. Qualitative Analysis

The mean pre-treatment gingival exposure ranged between 3.5 and 7.2 mm, reaching a reduction of up to 6 mm after infiltration with botulinum toxin at 12 weeks [26].

The amount of gingiva present in the smile was a parameter studied in all the studies, and its post-operative changes were carried out through photographs and/or video of the smile in all the research works. The method used frequently to evaluate the reduction in gummy smile after treatment was the one used by Polo in 2005 [27], using the upper right central incisor to calibrate the images of each patient from the highest point of the gingival zenith to the incisal edge [2,7,13,17,25,26,28,29]. However, other authors [3] took the upper lateral incisor as a reference, measuring the distance between its central cervical area to the lower part of the upper lip. Mazzuco and Hexsel, 2010 [28], in two of their four study groups, used the upper central incisors and in another, two upper premolar groups when evaluating a more posterior gummy smile. However, Sucupira and Abramovitz., 2012 [8] used only the upper central incisors; Suber et al. [30] and Dutra et al. [12] also used the upper central incisor, although its gingival measurement was the distance between the stomion point of the upper lip and the incisal edge of the upper central incisor subtracting the length from it. Gupta and Kohli, 2019 [17] did not mention their anatomic reference.

The muscles par excellence selected for blockade with BTX-A were: levator labii superioris, levator labii superioris ala nasalis, and zygomaticus minor; infiltrating from 1.25 to 7.5 units per side. Many of the studies [4,17,27,29] refer to the infiltration site as “Yonsei Point” described by Hwang et al. in 2009 [19] located in the center of the anatomical confluence of the three aforementioned muscles that form an imaginary triangle where its center would be the point of infiltration. Other authors [2,7,8,16,28] infiltrated only the levator labii superioris muscle.

The follow-up time of the patients in each study was analyzed to assess the surgical process over time; differentiating four stages: T0 (pre-surgical), T1 (post-surgical), T2 (first revision), T3 (second revision), T4 (third revision), T5 (fourth revision), T6 (fifth revision), and T7 (sixth revision). The most common follow-up times were at two weeks, one month, three and six months. The studies that took into account a longer follow-up time was that of Al Waily et al. [26] reaching 9 months, and Rajogoapal et al. [29] with a follow-up at T4 that ranged between 7 and 14 months, however, in this review, botulinum toxin was infiltrated again as the second cycle of treatment. There is a lack of follow up beyond 14 months of follow up in the included studies.

Nine studies provided information on the reduction in the gummy smile after infiltration with botulinum toxin at two weeks, representing a global sample of 213 patients. The decrease in the parameter is significant for all authors (*p* < 0.001) except Hexsel et al. [18] and Sucupira and Abramovitz [8], with mean values between −1.22 [31] and −5, 11 [25] millimeters, respectively.

Five studies are included in the meta-analysis estimates regarding the reduction in gummy smile 3 months after infiltration, grouping a total of 182 patients. The mean reduction difference ranges from 1 mm estimated by Somaiah et al. [13] to −5.95 mm in the work of Al Wayli et al. [26]. In all cases the reduction is significant (*p* < 0.001) except from Somaiah et al. [13].

### 3.3. Quantitative Analysis Results

A total of 11 studies evaluate the changes in millimeters of the gummy smile after infiltration with BTX-A at two weeks and three months.

To estimate the reduction in gummy smile at two weeks, 9 studies have been combined using a random effects model, estimating a reduction of −3.22 mm (95% CI between −4.43 and −2.01) (Figure 2). The heterogeneity between the included studies was high (Q Test = 170.04; *p* value < 0.05; I^2^ = 95.3%). The infiltrated botulinum toxin units after performing a meta-regression analysis does not affect significantly (*p* = 0.781) (Figure 3), with the number of millimeters that the gummy smile is reduced by the injected BTX-A units being independent.

Five studies that present results of the measurement of the gummy smile 3 months after the first infiltration with BTX-A have been combined, estimating a reduction of −2.70 mm (95% CI between −4.52 and −0.88) (Figure 4). Similar to the previous one, the meta-analysis shows heterogeneity (Q test = 193.35; *p* value < 0.05; I^2^ = 97.9%). The reduction in gummy smile at 3 months is not influenced by the number of infiltrated BTX-A units. The meta-regression analysis does not show a significant effect (*p* = 0.357) (Figure 5).

Through a subgroup analysis, no significant differences are found between the estimates of gummy smile reduction at two weeks and three months (Q test for intergroup heterogeneity = 0.221; *p* value = 0.638). Similarly, through a meta-regression in which the 11 studies are included, no significant effect of the follow-up time on the reduction in the gummy smile is found (*p* value = 0.630) (Figure 6).

### 3.4. Publication Bias

The two meta-analyses do not appear to be affected by publication bias. Using the Duval and Tweedie’s trim and fill method, no differences are observed in the estimation of the reduction between the meta-analyses at 2 weeks and 3 months with the observed studies and the meta-analyses with the inputted studies (Figure 7 and Figure 8). In addition, using the classic fail safe number, the number of non-significant studies necessary to include so that the significant estimate of the reduction ceased to be significant presented high values, specifically 1156 in the meta-analysis at two weeks and 530 at three months.

## 4. Discussion

The BTX-A infiltration is a novel technique to treat the over-gingiva exposure in non-growing patients [13,27]. Nowadays, we can find numerous publications in the scientific literature about the gingival exposure treatment with BTX-A, although only one is a systematic review [32] and another is a systematic review with meta-analysis [11]. Nevertheless, the gingival exposure reduction in the short term is not measured and neither are consequences of the BTX-A units per side infiltration in the medium or long-term. Since then, there have been multiple published research studies that meet the inclusion criteria in recent years [3,7,12,16,17,26,28,29]. The BTX-A topic as gingival over-exposure treatment succeeds in the dentistry community, especially for orthodontists who choose this technique to enhance the finishing of these type of cases.

If the cause of the gummy smile is the lip, if it is short, it could be treated by lip surgery, orthodontic intrusion, or BTX-A with or without hyaluronic acid. If there is hyperactivity in the upper lip, cheiloplasty or BTX-A with or without hyaluronic acid are indicated. If the origin of the increased gingival exposure is gingival hypertrophy, the therapeutic option would be gingivectomy, in the same way that if the origin is an altered passive eruption or another option, it could be coronal lengthening if there is an overeruption of the upper incisors.

Recent studies [10,21] defend fillers for the management of the gummy smile that involved the infiltration of hyaluronic acid in the pyriform fossa, achieving an average reduction of 1.37 mm of gingival exposure at 2 weeks [10]. The hyaluronic acid infiltrated in the skin area adds weight to the soft tissue and, therefore, can modify the gingival and incisal exposure. Infiltrated in the profile and/or in the vermilion, it can also modify the shape or volume of the red lip and, thus, also alter the exposure [33]. From clinical practice, we know that we can infiltrate the entire skin area of the upper lip, properly within the vermilion or in the “White Roll” (vermilion skin transition line), which is commonly known as “profiling” [33,34]. Hyaluronic acid without plumping the cutaneous part of the upper lip can beautify the final result and perhaps obtain a little exposure reduction.

The proper treatment for a severe gingival smile due to a vertical maxillary excess is the maxillary impaction. Even if, nowadays, the maxillofacial surgeons infra-correct the gingival smile as an aging prevention, as the tissue laxity aggravates with the age, the lips become longer and the exposure decreases in all patients [3,30]. The moderate gingival smile or the patients who refuse the surgery could be treated with a more conservative treatment by the BTX-A infiltration, as long as the patient agrees with the temporality and understands that it is not the gold standard treatment [3,25,28,30]. Just as with surgical infra-correction, to handle mild gingival smile by BTX-A infiltration is also one option to temporarily control the over exposure without aging consequences; as the aging tissue laxity progresses, most of the mild cases will show a proper quantity of gingival exposure.

To date, the available scientific literature is scarce, heterogeneous, and of low scientific quality. The indications for the technique are not well-defined and its use is not protocolized. Following the results of this systematic review and meta-analysis, a treatment algorithm is proposed based on the patient’s anterior gingival exposure (two to four millimeters and greater than four millimeters), depending on patient’s anatomical factors (Figure 9).

BTX-A decreases muscle strength in hyperactivity of certain muscles. The most common injection site corresponds to the anatomical confluence between the levator labii superioris muscle, and the levator labii superioris and ala nasalis muscle. Although multiple muscles are involved in the facial expression, this therapy is primarily indicated in patients with hyperactivity of these muscles. The detection of this target point requires a deep knowledge of the facial anatomy and a long learning curve, in order not to cause alterations in the symmetry of the smile and facial expression as mentioned in some studies [8,25,28,30]. Infiltrated toxin units differ considerably between studies: it ranged from less than two [7,8,13,25] up to five toxin units infiltrated per side [30]. A greater number of studies are required to determine the ideal dose, however, some articles defend a dose of 2.5 infiltrated toxin units per side [13,27,28]. If we study the meta-regression analyzes obtained in this research work, we can observe that the reduction in gingival exposure does not depend on the infiltrated units of BTX-A once the hyperactivity of the responsible muscle has been blocked.

The changes after infiltration are perceptible fifteen days after the start of treatment. From this moment until three months, there is total stability of the measurements. This solidity in the results is reflected in the meta-analysis by subgroups: the difference in mean reduction between both groups is −2.51 mm at two weeks and −2.24 mm at three months. Specifically in the perioral area, as it is highly mobile, the duration of the toxin decreases considerably if we compare it with other areas with less mobility in the upper third of the face.

The heterogeneity found in the present study tends to be convergent, and a significant reduction in the gummy smile with BTXA-A therapy is observed, albeit in different ranges. This could be due to the differences in the sample size of each study, and the absence of protocolization in this treatment.

At the present time, in the literature, there is no systematic clinical algorithm for the non-surgical treatment of gummy smile by infiltration of BTX-A. An increase in well-designed controlled prospective studies with a larger long-term sample size is required to reinforce the evidence of the protocol proposed in this research work.

*Limitations of the studies included.* There is a significant discrepancy in the follow-up time periods after the intervention. However, the studies with a review at six months [3,7,26,29,30,31] after infiltration with BTX-A showed a tendency to recurrence in the gummy smile. Research studies with a longer follow-up time, at eight [16] and nine months [26], found gingival exposure values similar to those studied before infiltration.

It is suggested to improve the risk of bias obtained in this research work that successive studies include a larger sample size, separating the results according to the etiology, location, and severity of the gummy smile; differentiating those that are male from female. In addition, it is recommended to establish a common study protocol: control of the patient’s posture at the time of photography and/or video, image calibration method, anatomical reference points, standardized review sequence, and to increase the follow-up time in the studies, analyzing the predictability from 6 months up to 12. All this would facilitate the study of the impact and importance of BTX-A in reducing the gummy smile.

*Limitations of the study.* The main limitation in our case was the significant discrepancy in the follow-up times periods after the intervention. Also, the high heterogeneity found between the studies although it is convergent. Research studies with a longer follow-up time analyzing the relapse of BTX-A are required.

## 5. Conclusions

The benefit of botulinum toxin in terms of improvement of gummy smile in the short term up to three months is demonstrated.

A significant reduction in gummy smile is estimated by BTX-A therapy two weeks after its application with a reduction of 3.22 mm. Its results gradually decrease over time, however, they stay satisfactory without having returned to their initial values at 3 months.The number of infiltrated BTX-A units does not influence the reduction in gummy smile obtained after treatment.

## Figures and Tables

**Figure 1 jcm-12-01433-f001:**
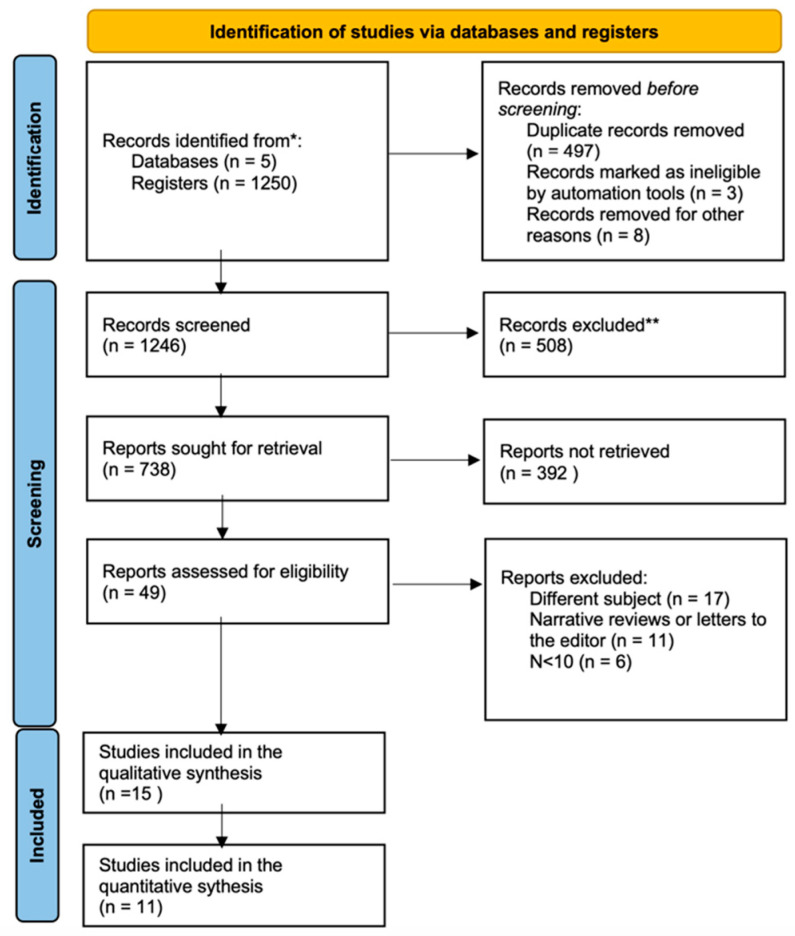
PRISMA 2020 flow diagram for new systematic reviews which included searches of databases and registers only [24]. (* Consider, if feasible to do so, reporting the number of records identified from each database or register searched (rather than the total number across all databases/registers). ** If automation tools were used, indicate how many records were excluded by a human and how many were excluded by automation tools).

**Figure 2 jcm-12-01433-f002:**
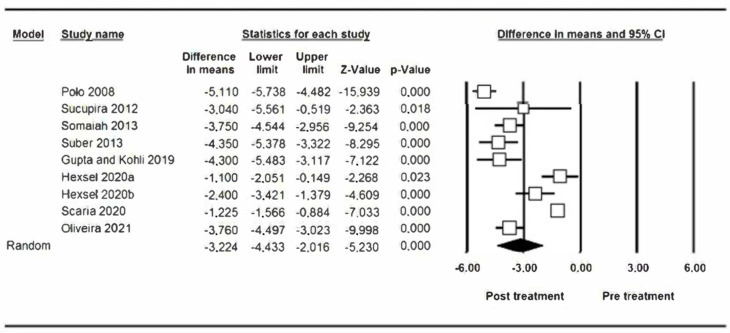
Reduction in gummy smile at two weeks.

**Figure 3 jcm-12-01433-f003:**
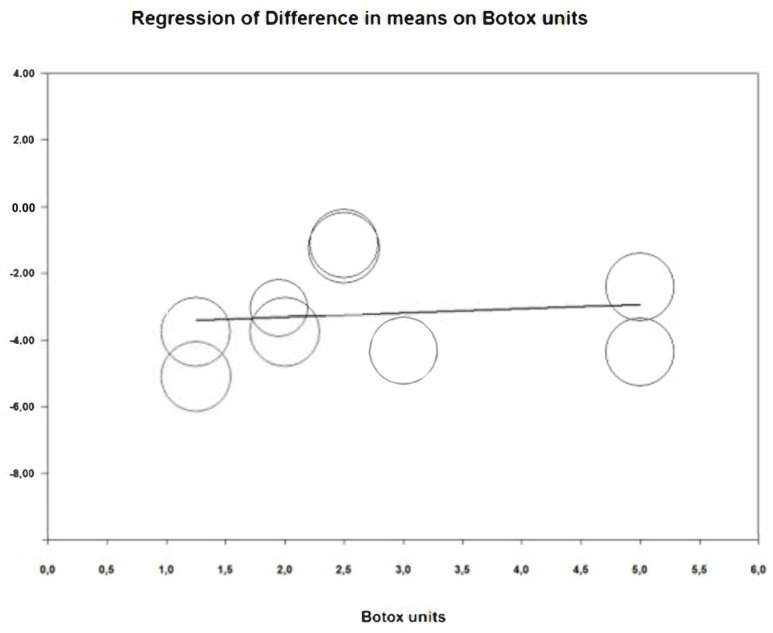
Meta-regression analysis units of BTX-A/gummy smile reduction at two weeks.

**Figure 4 jcm-12-01433-f004:**
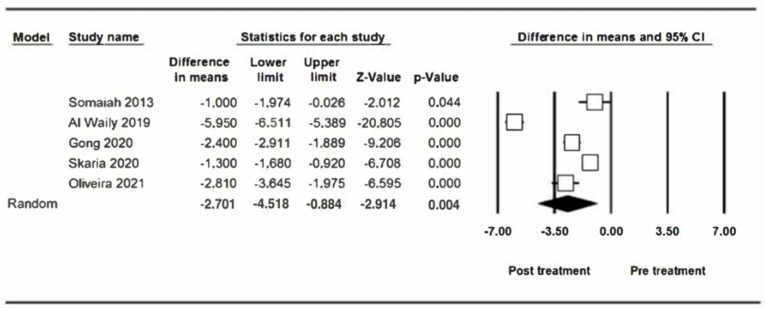
Reduction in gummy smile at three months.

**Figure 5 jcm-12-01433-f005:**
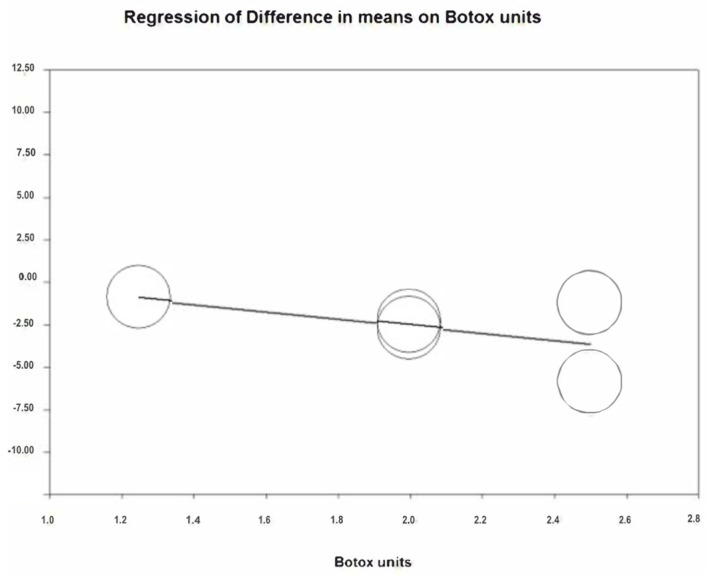
Meta-regression analysis units of BTX-A/gummy smile reduction at 3 months.

**Figure 6 jcm-12-01433-f006:**
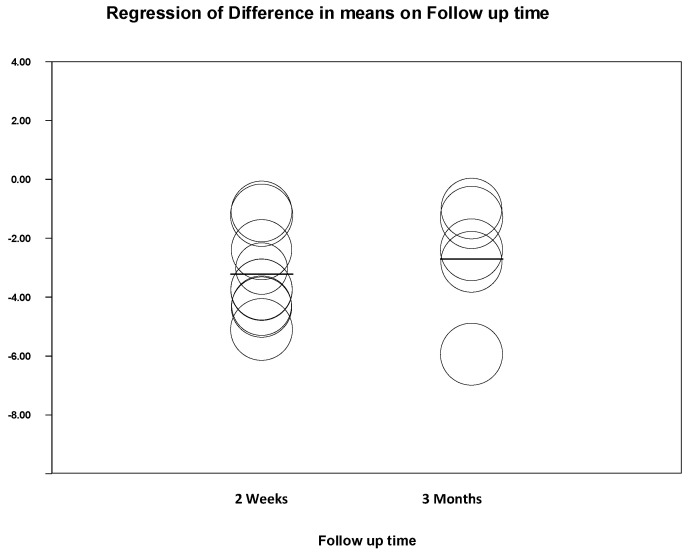
Subgroup meta-regression.

**Figure 7 jcm-12-01433-f007:**
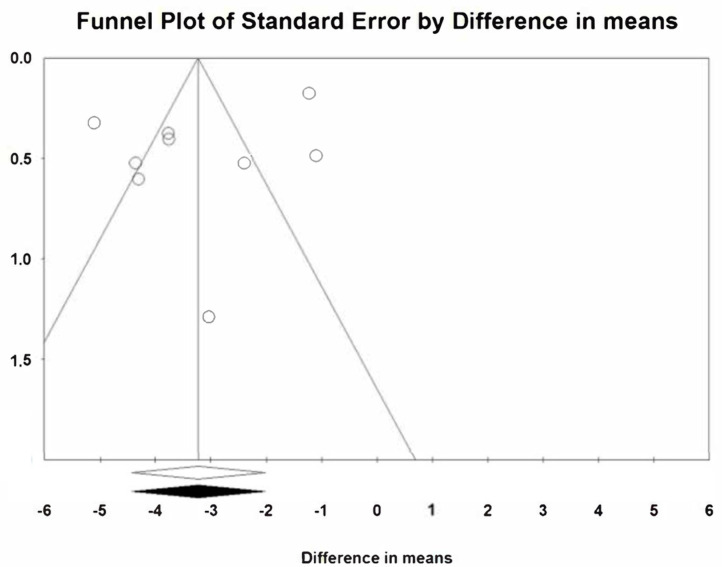
Publication bias at two weeks.

**Figure 8 jcm-12-01433-f008:**
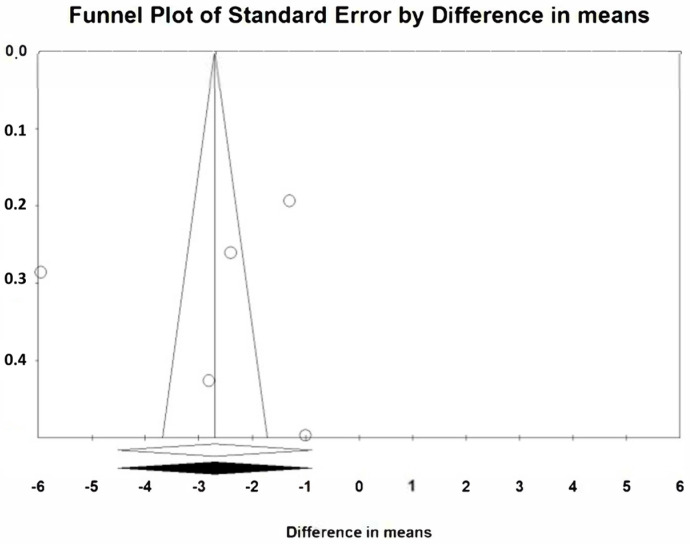
Publication bias at three months.

**Figure 9 jcm-12-01433-f009:**
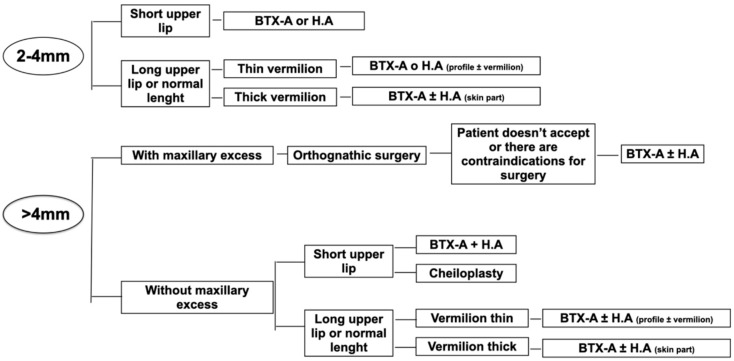
Proposal treatment algorithm according to anterior gingival exposure.

## Data Availability

Information is available upon request in accordance with relevant restrictions (e.g., privacy or ethical).

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
