# Peer review of "Non-Surgical Management of the Gingival Smile with Botulinum Toxin A—A Systematic Review and Meta-Analysis"

_jcm, 2023, doi:10.3390/jcm12041433_

Round 1

Reviewer 1 Report

The article is well written

I suggest to write about the (long lip) as a one of the complications of botox injection in the introduction section.

Reviewer 2 Report

Current manuscript, interesting, but some questions need to be clarified.

Abstract: I understand it unnecessary to say that something is statistically significant since significance is determined by statistics. It gets redundant!

In the last line, wouldn't it be better to say "stay satisfactory" rather than "stay satisfactorily"?

Introduction

2nd paragraph: starts and ends with smile. I suggest rewriting.

Page 2, 1st paragraph, 3rd line: when talking about the multiple ways to treat gummy smile, I suggest avoiding "..."

Materials and Methods

Page 2, subitem 2.2: ..."4 unvcontrolled descriptors:..." but cite more than 4.

Reading this section is sometimes difficult and confusing. If you can rewrite and improve it, making it easier for readers, that would be great!

Results

Page 4, subitem 3.1 and Figure 1: the numbers described from the beginning to the end do not fully correspond. Please review.

Page 5, 1st paragraph: you say that 92.3% of the included studies are at moderate risk of bias. So what, how to approach this and avoid problems? Also include in Discussion!

Discussion

Page 11: you say BTX-A infiltration is a "new technique" but reference #13 is from 1997. Is it really new?

You pointed out the limitations of the included studies, but what about your own study?

Conclusions

From my point of view, your 3rd conclusion is not adequate to the proposed study. I suggest including and commenting in the Discussion.

References

Of the 42 cited references, 11 (26.19%) are from 2009 or earlier, and of these 11 other 5 (45.45%) are from before 2000. I suggest updating and leaving only the essentials of these 11, if any. indispensable!

Reviewer 3 Report

Thank you for this meticulous screening of the literature and the solid report wth well presented results.

I would suggest, however, that you enhanced your warning in the discussion to be careful not to alter facial expression by the use of BTX (p13, line 312 ff).

Has this very important point not been mentioned in all of the cited studies ?    What is the use of diminishing a gingival smile at the cost of the natural facial expression ?

Reviewer 4 Report

Dear authors,
I evaluated the systematic study titled “NON-SURGICAL MANAGEMENT OF THE GINGIVAL 1 SMILE WITH BOTULINUM TOXIN A. A SYSTEMATIC RE- 2 VIEW AND META-ANALYSIS” with the following goal: “to analyze the effectiveness of BTX-A in the treatment of gummy smile in short and medium term, to study its stability and to evaluate potential complications”.
The topic is relevant but many concerns were raised.

ABSTRACT: it is poor and need to be completely redo.

INTRO
- line 47 “myotomy...”, please substitute the reticences
- lines 66-67: “duration ranges between 10 and 18 months. 21”, please, include more references to justify it
- line 76: “short and medium term”, please, define better what would be the mid term for the authors, justifying it.
- line 76: “analyzing its stability”, how the authors intended to check the stability through a systematic study? Clarify it, please.

M&M
- please, improve the section Inclusion and Exclusion criteria
- lines 125-126: “Cochrane Collaboration's ROBINS-I tool for Nonrandomized Studies of Interventions (NRSI).” How the authors considered to use this tool if they included RCT?

RESULTS
Fig. 1 has an “etc.” in the articles excluded. Please, insert whole information about the exclusions there
“Figure 2.- Reduction in gummy smile at two weeks.” The authors included info about 2wks and 3m of results. Please, where are the mid-term results?

- There are a lack of info for the studies included. Include information about the results up to 3 months ok! but and after 12 months and 3 years, e.g.?
- Do all included studies treatment patient and did the outcome at the same period (2wks and 3 months)?
- Extreme heterogeneity was observed among the included studies

DISCUSSION
- unnecessary table included (remove table 1 and describe in the text)
- the authors included a “Proposal treatment algorithm” in the discussion. Please, review completely the discussion section

CONCLUSION
- it is wrong. Please, rewrite it.

Round 2

Reviewer 2 Report

In the attached pdf there are several marks and questions that still need to be answered.

Author Response

Please, see the attachment. Thank you. 

Reviewer 4 Report

Dear authors,
I evaluated the systematic study titled “NON-SURGICAL MANAGEMENT OF THE GINGIVAL SMILE WITH BOTULINUM TOXIN A. A SYSTEMATIC REVIEW AND META-ANALYSIS” with the following goal: “to analyze the effectiveness of BTX-A in the treatment of gummy smile in short and medium term, to study its stability and to evaluate potential complications”.
The topic is relevant but many concerns were yet raised.

ABSTRACT: it needs to be improved yet (results)

- The authors included info about 2wks and 3m of results. Where are the mid-term results? It would be important to show for the patients the predictability of the results after 6 and 12m.
- Lines 85-86: “The objective of this study is to estimate the improvement of the gummy smile with BTX-A in the short and medium term” - there is no mid-term in this article.

- Did all studies treat the patient and outcome at the same period (2wks and 3 months)?
- Extreme heterogeneity was observed among the included studies

DISCUSSION
- unnecessary table included (remove table 1 and describe in the text)
- the authors kept a “Proposal treatment algorithm” in the discussion. Please, review it

CONCLUSION
- there is mistakes. Review it.
